# Establishment and Social Impacts of the Red Imported Fire Ant, *Solenopsis invicta*, (Hymenoptera: Formicidae) in Taiwan

**DOI:** 10.3390/ijerph18105055

**Published:** 2021-05-11

**Authors:** Yu-Sheng Liu, Sin-An Huang, I-Lin Lin, Chung-Chi Lin, Hung-Kuan Lai, Chun-Hsiang Yang, Rong-Nan Huang

**Affiliations:** 1Department of Business Management, National Sun Yat-sen University, Kaohsiung 80424, Taiwan; yuseing@g-mail.nsysu.edu.tw; 2National Red Imported Fire Ant Control Center, Council of Agriculture, Executive Yuan, Taipei 10617, Taiwan; rivenser@gmail.com (S.-A.H.); cclin@cc.ncue.edu.tw (C.-C.L.); aqualai@outlook.com (H.-K.L.); Yshang.yang@gmail.com (C.-H.Y.); 3Department of Political Science, National Taiwan University, Taipei 10617, Taiwan; elimlin36@gmail.com; 4Department of Biology, National Changhua University of Education, Changhua City 50007, Taiwan; 5Department of Entomology, National Taiwan University, Taipei 10617, Taiwan

**Keywords:** *Solenopsis invicta*, health effect, urbanization

## Abstract

The first report of the red imported fire ant (RIFA), *Solenopsis invicta* Buren, in Taiwan was in the city of Taoyuan in 2003. The government has made great efforts to bring RIFA-infested areas under control. RIFA has gradually spread outward since its discovery, but it is still confined in northern Taiwan, in part due to the control efforts. RIFA is well established in densely populated environments (i.e., urban areas), causing damage to public utilities and significantly affects the inhabitants of Taiwan. Out of 10,127 human encounters with RIFA reported by the Plant Pest Information Management System in the Bureau of Animal and Plant Health Inspection and Quarantine, Council of Agriculture, Executive Yuan, 3819 (37.71%) persons were stung, with 834 (21.8%) persons exhibiting wheal-and-flare reaction (swelling and redness of the skin). Among the victims, 288 (7.5%) sought medical care, and about 21 (0.6%) developed severe cellulitis and urticaria. Unexpectedly, 2.8% (106) of the victims exhibited anaphylactic shock, which was higher than previously reported cases (1%). The high anaphylactic shock percentage was probably because most victims were elderly farmers or because Asian people have higher sensitivity to the RIFA sting. RIFA is well adapted to the environmental conditions in Taiwan, which makes it extremely difficult (if not impossible) to eradicate. The management of RIFA in the future should focus on lowering the speed of spread to mitigate possible dangers to the inhabitants. Six major challenges of RIFA management in Taiwan are also discussed.

## 1. Introduction

The red imported fire ant (RIFA), *Solenopsis invicta* Buren, is classified as one of the major 100 most invasive ants worldwide. It was introduced into the United States from its native habitat in Brazil during the 1930s. After colonizing the US for around 80 years, the RIFA spread across the Pacific Ocean to Australia, China, and Taiwan from 2001–2004 [1,2]. Unexpectedly, the RIFA invasions were almost simultaneously detected in Japan and Korea in 2017 [3]. The transnational spread of RIFA has become more rapid, posing an increased threat of its invasion. The increased speed with which RIFA has been spreading probably results from an expansion in global trade [2,4].

Outside its native habitat, RIFA poses a serious threat to the ecological balance owing to its extremely aggressive characteristics, as it out-competes native ant species. RIFA is also one of the more threatening invasive species for human health due to its venomous sting [5]. RIFA is more threatening than bees or wasps, even though its size is much smaller. A large mature RIFA colony may have 200,000–300,000 workers that are very aggressive once their nests are disturbed [5]. If provoked, they attack in swarms and each RIFA individual can sting a human repeatedly without breaking its stinger, unlike bees in which the stinger breaks and remains in the victim’s skin. Therefore, the RIFA sting causes more serious damage than the bee sting [6].

The RIFA sting immediately causes intense itching or burning pain that is usually short-lived, lasting from a few seconds to a few minutes. The symptoms are followed by a mild or intense itching or burning feeling and the development of pus-filled blisters, usually within 30 min of a fire ant attack [7,8]. Because fire ants tend to attack their victims in groups, the stings often occur in clusters that set them apart from other insect stings. Although most stings heal on their own without treatment, the itching tends to become stronger over the next few days and blisters appear like pimples that could last for a month. If scratched or broken, the blisters could worsen, for example, because of secondary infection [6]. Anaphylactic shock and even death can also occur in a few sensitive individuals [9,10,11].

To eradicate RIFA, the National Red Import Fire Ant Control Center (NRIFACC) was launched in November 2004 in Taiwan. The operational aspects of the control center were based on four strategies: public education, surveillance (or monitoring), treatment (mainly insect growth regulator (IGR): pyriproxyfen and methoprene) and movement control. Accordingly, a Plant Pest Information Management System of Taiwan (https://phis.baphiq.gov.tw(accessed on 8 July 2020)) was established to provide a platform for the consolidation of RIFA information on notification, detection, and prevention [12]. The platform was also launched in 2004 by the Bureau of Animal and Plant Health Inspection and Quarantine (BAPHIQ), Council of Agriculture of the Executive Yuan, and covered information from 2004 to the present. The government agencies in Taiwan have invested considerable efforts to control and prevent the establishment of RIFA since 2004, nonetheless, it has not been effectively eradicated so far. Its invasion not only endangers the ecological balance in the established area, but also destroys various facilities and imposes significant threats to human health owing to its aggressive nature. The threat of RIFA to the residents of Taiwan would be more serious than to those in other countries because Taiwan is densely populated with limited land (651 persons/km^2^, top 9 globally), this is especially true in heavily RIFA-infested areas like Taoyuan, Taipei, and New Taipei City, which are densely populated urban environments with highly developed industries and commercial activities. For instance, the population in the RIFA-infested area accounts for 38% of the total population in Taiwan (https://www.ris.gov.tw/app/en/3910 (accessed on 19 April 2021)). Therefore, in addition to destroying the ecological environment and endangering public facilities [5], the damage caused by RIFA to the general public in Taiwan deserves closer attention. Taiwan is one of the first countries in Asia to be invaded by RIFA, but the effect of RIFA on the health of people in the Asian region has rarely been reported. Therefore, this report provides an overview of the effect and characteristics of the spread, establishment and impacts of RIFA in Taiwan. Moreover, the challenge of RIFA management in Taiwan is also addressed.

## 2. Materials and Methods

For this study, a total of 34,131 observations were retrieved from the Plant Pest Information Management System of Taiwan on 20 March 2020. Among the 34,131 cases, 27,279 (79.92%) were at the time under treatment and control, 7 (0.02%) were in notification (cases have been reported to the authorities), 1958 (5.74%) were determined to be caused by non-invasive ants, and 4887 (14.32%) cases were closed, meaning the RIFA were eradicated and can no longer be found. These data included statistics on whether any people were stung by fire ants, whether they sought medical attention, and symptoms after the sting, such as anaphylactic shock, cellulitis, or urticaria.

The statistical analyses were based on the number of persons who were stung by the RIFA and had their relevant medical treatments and symptoms recorded in the Plant Pest Information Management System. We applied cross-tabling to evaluate the relationship between seeking medical care and developing symptoms, such as wheal-and-flare reaction, anaphylactic shock, cellulitis or urticaria.

The distribution of the RIFA in Taiwan was graphed according to the data based on toll free hotline reports from residents (passive) and active surveillance by NRIFACC annually. The toll-free hotline cases reported were further confirmed on-site by NRIFACC members. Active surveillance was performed annually using potato chips to attract foraging ants, as described by Bao [13,14].

## 3. Results and Discussion

### 3.1. Chronological Distribution of the Red Imported Fire Ant (RIFA) in Northern Taiwan

RIFA was simultaneously discovered in northern (Taoyuan City) and southern (Chia-Yi County) parts of Taiwan in 2003 [15]. Figure 1 shows the chronological expansion of RIFA in northern Taiwan from 2004 to 2020. Before 2009, RIFA was effectively contained in Taoyuan City because of the initial National Three-Year Action Plan (2005–2007) for the control and eradication of RIFA. The control strategies in the plan included a large-scale broadcast of IGR-based bait in the areas invaded by RIFA in combination with strengthened public education, intensive surveillance (or monitoring) in high-risk areas, and strict control of human-assisted dispersal (movement control). However, the strategies for the prevention and control of RIFA were modified for hot spots and high-risk zone treatment owing to budget constraints after 2012 [13]. This modification accelerated the spread of RIFA southward and northward from Taoyuan City (Figure 1).

In addition to Taoyuan and Chia-Yi County, RIFA has also been sporadically reported from Taichung in central Taiwan (2011) and Yi-Lan in northeast Taiwan (2007, 2014, 2019). The initial goal of NRIFACC was to eradicate RIFA from Taiwan completely, and the RIFA control practices have been consecutively performed for 16 years in Taiwan. Although the eradication program is still far from being successful, complete eradication has been achieved in Yi-Lan, Taichung, and Chia-Yi counties [13]. RIFA was also found on Kinmen Island four years ago (2015); however, genetic data suggested that it was derived from China, and not from Taiwan [unpublished results].

RIFA queens have been reported to disperse at least 12 miles (19.3 km) away during nuptial flights along with the wind [16,17,18]. RIFA was discovered in Australia in 2001; Scanlan’s models predicted that *S. invicta* would infest 763,000–4,066,000 km^2^ by the year 2035 (ca. at the rate of 25,000–130,000 km^2^/year) and would be found at 200 separate locations around Australia by 2017–2027, depending on the rate of spread, if no control measures were taken [19]. Moreover, RIFA appears to have invaded the United States through Mobile, Ala, in the early 20th century and the US Department of Agriculture estimates that it has expanded westward approximately 120 miles (193 km) per year [20]. At this rate, RIFA would infest all of Taiwan within two years, because the area of Taiwan is only 36,000 km^2^, its length from the north to the south being 394 km. Moreover, the whole island of Taiwan is warm and humid, providing highly suitable habitat for RIFA. The invasion of RIFA could even occur over hundreds of kilometers by anthropogenic movements, such as potted plant and turf transportation [5]. Although RIFA has spread out slightly northward and southward from Taoyuan County over time (Figure 1), it is still contained in northern Taiwan and has not aggressively infested the whole island since its initial discovery. This indicates that the control practice for RIFA in Taiwan is to some extent effective in limiting the significant spread of RIFA, which, otherwise, spreads quickly year-on-year in natural environments [19].

### 3.2. Health Effects of RIFA on the Residents of Taiwan

RIFA was first reported in Australia, China and Taiwan at the beginning of the 21^st^ century [1] and is a new pest for Asians. The RIFA venom contains mainly alkaloids and proteins [21]. The sting of RIFA can cause an extremely unpleasant burning pain in humans, and the severity of RIFA sting reactions ranges from typical local reactions to systemic allergic ones [22]. People vary in their sensitivity to fire ant stings, and those hypersensitive to their venom might develop fatal anaphylactic reactions [23]. Local reactions are generally classified into three types: wheal-and-flare reactions, sterile pustule reactions, and extensive local reactions.

The information on the Plant Pest Information Management System included 10,127 reports of people who encountered RIFA (Table 1). Of these, 6308 (62.29%) were not stung, and 3819 (37.71%) were stung. Comparing this to the number of Americans stung by RIFA is difficult. About 40 million people in the southeastern United States live in areas invaded by RIFA and approximately 14 million people (35%) there are stung by fire ants annually [24,25], however the number of encounters that lead to stings cannot currently be compared, and likely varies greatly with location. According to the Scripps Howard Texas Poll (March 2000), 79% of Texans had experienced the sting of RIFA in the year of the survey, whereas 20% of Texans reported that they had never been stung. Their detailed survey showed that West Texans were least likely to have been stung by RIFA (61%) in contrast to 90% in the central area, 89% in the east, 86% in the Gulf, 78% in the South, and 72% in North Texas [25]. This may be closely related to the density of fire ants and human activities in various regions. Regardless, the existence of RIFA does put people at risk of being stung.

The present study showed that all of the Taiwanese people stung by RIFA, only 21.00% (802 people) developed skin redness and swelling, which are the most common reactions to RIFA sting. It is a reaction in which the blood serum enters the surrounding tissues because of vasodilation [7], typically observed within 30 min to 1 h [26]. However, the proportion of people in Taiwan who develop redness and swelling after being stung by RIFA was much lower than reported previously. In the literature, almost all people stung by RIFA are reported to have experienced skin redness and swelling, and most (85–96%) further developed the RIFA-specific white pustules that last for 48 to 72 h [7,8,11,27].

Of the Taiwanese stung by RIFA, approximately 7.54% (288) sought medical treatment (Table 1), which is much higher than the percentage reported in the literature. In earlier studies, such as Clemmer’s telephone survey [28], approximately 2% of those stung by RIFA sought medical assistance. According to Caldwell et al. [29], only 0.94% of people in California visit hospitals or clinics, of which 0.02% develop severe allergies (anaphylaxis). This is far lower than the proportion of people in Taiwan who develop allergic reactions after RIFA stings. Of the 288 Taiwanese who sought medical attention for RIFA stings, approximately 0.55% had cellulitis or urticaria (21 people), and 2.78% had anaphylactic shock (106 people). The status of RIFA stings, the associated symptomology, and the status of seeking medical care among patients are presented in Appendix A.

RIFA also frequently occurs in agricultural areas and has a relatively severe effect on Taiwanese farmers because Taiwan has adopted intensive farming and some farmers often undertake agricultural practices barefoot, which makes them more vulnerable to RIFA attacks (Figure 2a,b). Owing to its aggressive character, RIFA can sting anyone. For instance, Figure 2c shows a baby stung by RIFA. Summer time is very hot in Taiwan, and the residents usually put a blanket on the ground indoors for babies to sleep on at noon. In this case, RIFA invaded the house and stung the baby. Therefore, RIFA not only colonizes the wild but also invades houses. Individuals with a severe reaction to being stung by RIFA cannot go to work or school. Thus, fire ant stings significantly affect daily life and national productivity.

Because of the large area affected by RIFA in the United States, the cost of prevention and control has become unaffordable for the government agencies. Government agencies no longer carry out official broad-scale prevention and control, and the residents have to control RIFA on their own [30]. Texas was the most heavily infested state in the United States. According to an estimate in 1998, each household in the Texas metropolitan area spends US$150.79 on fire ant control every year, of which US$9.4 was used for medical expenses incurred for the treatment of RIFA stings. Nearly 9% (US$47.1 million) of the total medical expenses of US$526 million in Texas are related to RIFA stings [31]. Taiwan is located in a subtropical zone with high humidity and mild temperatures, which are most favorable for the survival of RIFA. However, Taiwan is a small country that cannot withstand RIFA infestation throughout the island. In Taiwan, RIFA is still categorized as a plant-specific pest and disease for which all control measures are governed by public agencies. Citizens do not have to spend their own money on the prevention and control of RIFA. If RIFA spreads to all the counties in Taiwan, it will become a general plant disease and an insect pest. In that case, it will pose a significantly greater threat to the general public, leading to an increased frequency of people seeking medical treatment for RIFA stings. According to the willing-to-pay cost for avoiding RIFA in households (42 USD/year) and agriculture (42 USD/year) sector, the benefit of a RIFA controlling policy in the major infested city (Taoyuan) was estimated to be 80 million annually [32] in Taiwan.

### 3.3. Urbanization of RIFA in Taiwan

The RIFA population in Taiwan also has been established in highly urbanized areas with intensive human activities (Figure 3). Figure 3a,b show that RIFA frequently colonizes the rooftop gardens in tall buildings and even the drain pipes in these gardens (Figure 3b). The RIFA colonies block the drainage during the rainy season and can also enter houses through the pipe during cold winters. RIFA colonies are also frequently found in playgrounds (Figure 3c) or pathways (Figure 3d) along waterways, which are the main areas for outdoor activities in the Taipei and Taoyuan metropolitan areas. People frequenting such areas are at high risk for RIFA stings if the playgrounds are not freed of the ants. Figure 3e,f show RIFA nests located in a Athletes’ village for the XXIX Summer Universiade, 2017, Taipei, Taiwan [33]. At the end of the Summer Universiade, the village was remodeled into public housing units for city residents. Two large RIFA colonies were found in the courtyard surrounding the public housing unit (Figure 3e,f).

Urbanization has been accelerating over the past few centuries [34]. It is estimated that more than two-thirds of the world’s population will migrate from rural to urban areas by 2050, increasingly in highly dense cities [35]. The population of Taiwan is projected to be 23,561,236 people in the mid-2020s, according to United Nations data (Appendix A). Although it accounts for 0.31% of the total world population and ranks 57 in the list of countries (and dependencies) by population, 78.9% of the population lives in urban areas. Moreover, 40% of the Taiwanese population is distributed in northern Taiwan (Appendix A), which is the area most infested with RIFA. RIFA is likely to have a more significant effect in urban settings, particularly in Taiwan.

### 3.4. Destruction of Electrical Equipment by RIFA

RIFA is attracted by electricity [36,37], so it can cause expensive damage to sensitive electrical equipment, such as electrical utilities at airports, high-speed trains, or electric stations. RIFA nests were detected in a cable box in Taoyuan International Airport in 2004 (Figure 4a,b), the year after RIFA was first found in Taiwan. In 2014, RIFA also invaded the Songshan Airport (Figure 4c,d) in Taipei city. Many of RIFA workers were found inside a runway lamp during repairs (Figure 4c) and a large mound of their colonies was present beside the lamp (Figure 4d), indicating that the failure of the lamp was due to invasion by RIFA. After extensive surveillance at the Songshan Airport, over 200 RIFA colonies were found on about 80 hectares of the runway. The runway lamp is a critical signal for the takeoff and landing of airplanes and damage to it could result in serious safety problems. Control of RIFA at the airport is difficult because of the frequent airplane operations and can only be conducted at midnight or when there are no airplanes on the runway. However, at the Songshan Airport, RIFA was wholly eradicated with a 2-year extensive control regimen using a two-step method (broadcasting a bait insecticide once or twice a year followed by individual mound treatments) [38]. Figure 4e shows a circuit breaker in a resident’s house invaded by RIFA. Such an invasion frequently causes short circuits or even fires.

### 3.5. Challenges of RIFA Management in Taiwan

RIFA is a major aggressive invasive ant species worldwide and has been categorized as a “super pest” owing to its expansion rate and severe effects on the nation’s economy, health, ecology, and lifestyle [39,40,41]. Despite great efforts, the United States has failed to eliminate RIFA, in part due to the lack of experience and absence of effective natural enemies [30]. Currently, RIFA has intensively invaded more than 13 states in the southeastern United States; it has even spread northward to Virginia and westward to California [20]. Over 40 million people are threatened by the RIFA invasion in the United States, and approximately 14 million people (35% of the total population) have experienced the sting of RIFA [24,25]. Although Taiwan’s government continues to invest significant endeavors to prevent its spread southward, there are multiple major challenges to RIFA eradication:(1)RIFA is likely to have been established in Taiwan longer than previously reported. Earlier detection of an exotic organism is the general principle of biosecurity and offers a better chance of eradication. For instance, New Zealand has successfully eradicated RIFA because three incursions were detected at ports of entry very early, with only a few colonies [3]. In contrast, RIFA was thought to have arrived in 1933 in the United States but was not discovered until 1942 [42]. The cases in Taiwan, China, and Australia were similarly discovered late [41]. RIFA was likely present in Taiwan for more than five years before it was discovered in 2003 [3].(2)Highly reproductive queens can fly distances of more than 10–20 km and there are difficulties in detecting incipient infestation [20]. The nuptial flight of imported fire ants is believed to be the most successful strategy in terms of population expansion, frequently occurring in light winds of up to 5 miles/h (8.05 km/h) and temperatures between 23.8–33 °C [43]. Although 99% of the mated queens fly less than 1 mile (1.61 km) from the point of origin of their flight, there is evidence that in extreme cases, queens can fly or can be carried by the wind for 7–10 miles (11.27–16.09 km). In the United States, the most intense and extensive flights occur in May, June, and July; in Taiwan, however, they can occur throughout the year because of the warm climate and high humidity. This facilitates the rapid expansion of RIFA and renders it more difficult to manage. The current monitoring system for RIFA in Taiwan employs potato chips to attract the foraging ants [14], but it is still not easy to detect an incipient infestation. Dogs are highly sensitive agents for RIFA monitoring in areas that contain a low density of fire ant nests; however, their cost is too high and unaffordable for large-scale applications [44].(3)Movement control is difficult to achieve. Human-assisted dispersal through nursery pots and soil is the most challenging part of controlling RIFA because it can easily transport the ants over a long distance [41,45], as evidenced in the case of Taichung World Flora Exposition on 2 November 2018, in Taiwan [13]. The dilemma has to be overcome by promulgating regulations. For example, in Japan an Invasive Alien Species Act (IAS Act) was adopted in June 2004 and put into force from June 2005 with the aim of preventing the adverse effects of Invasive Alien Species (IAS) on ecosystems, human safety, agriculture, forestry or fisheries. In Taiwan, RIFA is currently governed only by the Plant Protection and Quarantine Act.(4)Manpower and the companies involved in the broadcast baits for RIFA control are insufficient, and the turnover rate is too high. The management of RIFA in Taiwan is coordinated by BAPHIQ; however, the person responsible for RIFA control is an entry-level employee in the local government or township. Unfortunately, the turnover of entry-level employees is high and it is not easy for them to become familiarized with the RIFA Control Act.(5)Meteorological conditions disfavor bait application in Taiwan. Currently, several synthetic insecticides are used as active ingredients in fire ant baits. In particular, baits with two IGR (pyriproxyfen and methoprene) are the main active ingredients for RIFA control in Taiwan. However, the phagostimulant components in the baits are quite sensitive to humidity. Taiwan is an island country with a tropical and subtropical climate, and it is often rainy in the northern part of Taiwan. According to the Central Weather Bureau of Taiwan, the number of days of precipitation in Taipei is approximately 170 days/year, which means that almost every other day is a rainy day (Appendix A). These conditions make it quite challenging to broadcast bait applications for RIFA control. To operate effectively in a highly humid environment, a new fire ant bait base carrier for moist conditions was developed in 2010 [46].(6)Residents are reluctant to change their faulty concepts regarding the control of RIFA. RIFA is a social insect with the queen(s) as the only female with fully developed ovaries and responsibility for the reproductivity of the whole colony. Such insects can only be controlled by eradicating the queen(s), which usually hides inside the nest. The IGR-based bait used for RIFA control can wipe out the entire colony by destroying the queen(s) through trophallaxis. Despite repeated advocacy education programs, Taiwanese residents still treat RIFA by imitating the concepts used for the control of mosquito or other agricultural pests, which aim to kill all visible individuals with contact insecticides.

## 4. Conclusions

The RIFA-infested area in Taiwan has gradually increased since its discovery, and the eradication of RIFA is probably unfeasible, unless the challenges mentioned above can be overcome. However, we could slow down its spread southward to the main agricultural sectors of Taiwan, which are more favorable for RIFA colonization. This would require sufficient financial support from the government and public awareness to facilitate participation in RIFA management. Finally, a fully responsible coordinating agent is an urgent requirement, because the lack of coordination hampers timing management in different areas and leads to treatment failure due to re-infestation from neighboring untreated infestations [3].

## Figures and Tables

**Figure 1 ijerph-18-05055-f001:**
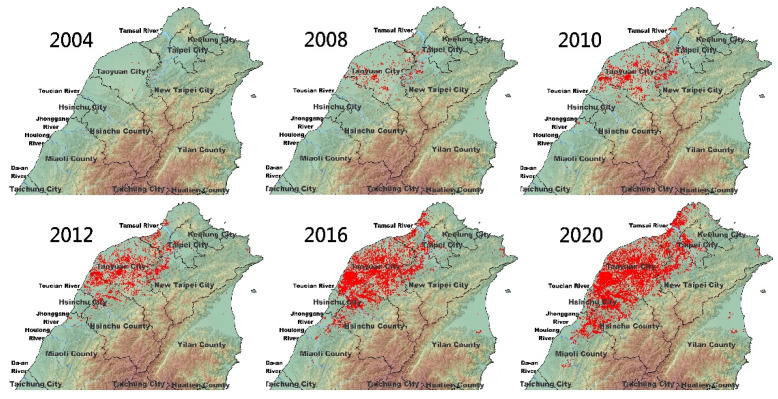
Chronological distribution of red imported fire ant (RIFA) in northern Taiwan. The localization of RIFA was surveyed using potato chip annually or was reported by the general public (● invaded area).

**Figure 2 ijerph-18-05055-f002:**
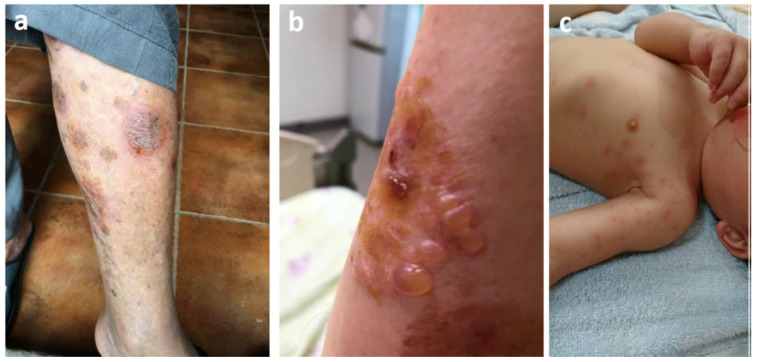
Health impacts of red imported fire ant (RIFA) on the residents of Taiwan. (**a**,**b**). A farmer’s leg and arm stung by RIFA during agricultural practice in the field; (**c**). A child stung by RIFA during a nap on the ground inside the house.

**Figure 3 ijerph-18-05055-f003:**
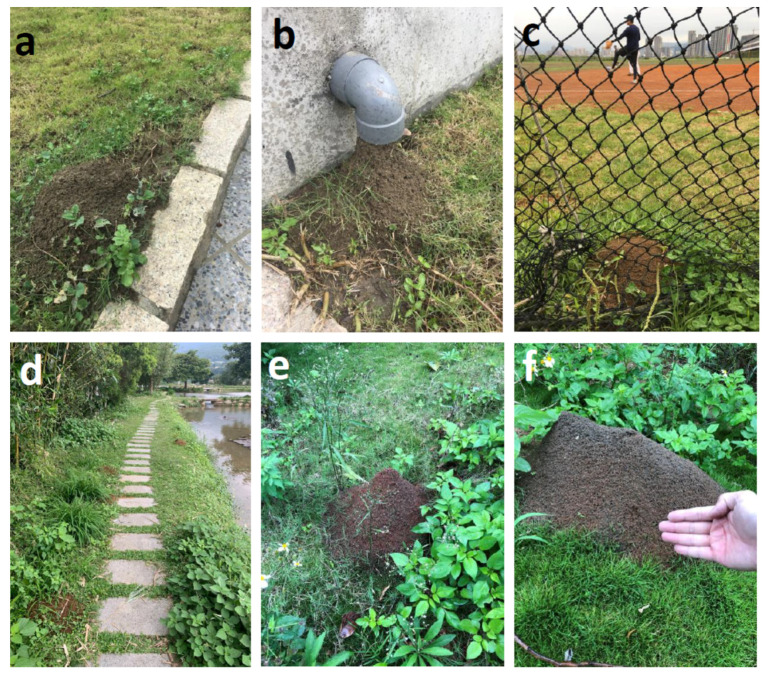
Urbanization of red imported fire ant (RIFA) in Taiwan. (**a**). RIFA colony in a rooftop garden. (**b**). RIFA colonizing in a drain pipe in a rooftop garden. (**c**,**d**). RIFA colony in a playground and pathway located beside a waterways. (**e**,**f**). RIFA colony located in a player village for the XXIX Summer Universiade, 2017.

**Figure 4 ijerph-18-05055-f004:**
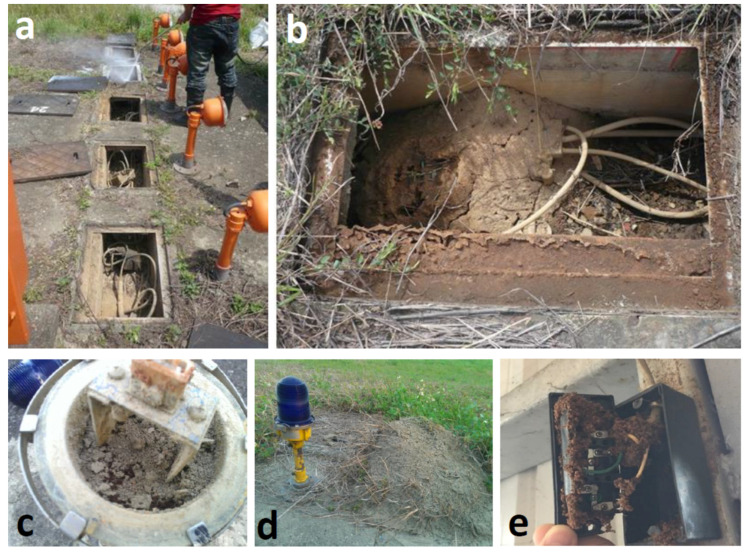
RIFA destruction to electric facilities in Taiwan. (**a**,**b**). RIFA colonized in cable boxes in Taoyuan International Airport. (**c**,**d**). RIFA invaded into a runway lamp (**c**) and a colony located beside a runway lamp (**d**) at Songshan Airport. (**e**). RIFA invaded a circuit breaker in a resident house.

**Table 1 ijerph-18-05055-t001:** Association of seeking medical care and symptomology within people having been stung by the red imported fire ant (RIFA).

			Seeking Medical Care (Percentage)			
Characteristics	Total Record (%)		No (%)		Yes (%)	
**Wheal-and-flare reaction**						
No	3017	(79.0%)	2835	(94.0%)	182	(6.00%)
Yes	802	(21.0%)	696	(86.8%)	106	(13.2%)
**Anaphylactic shock**						
No	3713	(97.2%)	3495	(94.1%)	218	(5.90%)
Yes	106	(2.8%)	36	(34.0%)	70	(66.0%)
**Cellulitis or Urticaria**						
No	3798	(99.5%)	3510	(62.40%)	288	(37.6%)
Yes	21	(0.50%)	21	(100.0%)	0	(0.00%)
Number of People Stung by RIFA	3819		3531	(92.5%)	288	(7.50%)

## Data Availability

The RIFA information on notification, detection, and prevention of Taiwan’s animal and plant epidemics in Taiwan can be found at https://phis.baphiq.gov.tw. accessed on 8 July 2020.

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
