# Peer review of "Establishment and Social Impacts of the Red Imported Fire Ant, Solenopsis invicta, (Hymenoptera: Formicidae) in Taiwan"

_ijerph, 2021, doi:10.3390/ijerph18105055_

Round 1
Reviewer 1 Report
This paper reports on the occurrence of Solenopsis invicta in Taiwan and their impacts on human beings. My major problem with this manuscript is that there a lack of references to support assertions. For instance, Lines 70-76, 143-145. The language needs to be polished. Otherwise, the manuscript is well prepared. I have only minor comments.
Lines 79-81: Repetition of previous paragraphs.
Line 230: China mainland is 2004.
Figure 1: There is no need to give same title for each map. The year can be placed in the left-up corner or other place of each map. Annotation of red dot can be placed in the figure legend.
Author Response
Dear Reviewer
We have carefully addressed the questions you raised.
please see the attached file.

Reviewer 2 Report
I reviewed the manuscript ijerph-1192811 entitled “The adaptation and impacts of the red imported fire ant, Solenopsis invicta, (Hymenoptera: Formicidae) in Taiwan”, submitted for publication in IJERPH by Liu et al.
I think the topic is interesting. Estimating the health consequences of an invasive species such as S. invicta can help realize the urgency to limit its spread.
However, I have several concerns about this manuscript:
- There is a systematic lack of citations. It is totally unacceptable for a scientific paper to not cite its sources!! The authors MUST correct this major issue.
- The results and discussion section is way too long! And the main results should be at the beginning! Not at the end! I think you should concentrate on your two main results: 1) rapid spread and 2) lots of health consequences. All other discussion (i.e., sections 3.2, 3.3 and 3.5) are interesting but not the real topic of the paper and should be reduce to the maximum or removed.
- This paper is not about adaptation (not in the ecological and evolutionary meaning of it). I recommend changing the title to something like: “Human health consequences of the rapid spread of the red imported fire ant, Solenopsis invicta, in Taiwan”. And removing the term “adaptation” throughout the manuscript.
Specific comments:
Abstract (my corrections in red)
The red imported fire ant (RIFA), Solenopsis invicta Buren, was discovered in 2003 in the city of Taoyuan, Taiwan. The government has made great efforts to reduce RIFA-infested areas. RIFA gradually spread outward since its discovery, but it is still confined in northern Tai-wan, in part due to the control efforts. RIFA is well adapted to densely populated environments (i.e., urban areas) and thus, causes damage to public facilities and significantly affects the inhabitants of Taiwan. According to 10,127 records from the Plant Epidemic Information Network in the Bureau of Animal and Plant Health Inspection and Quarantine, Council of Agriculture, Executive Yuan, 3,819 (37.7%) persons have already been stung by RIFA, with 834 (21.8 %) persons exhibiting wheal-and-flare reaction (swelling and redness of the skin). Among the victims, 288 (7.5%) sought medical care, and about 21 (0.6%) developed severe cellulitis and urticaria. Unexpectedly, 2.8% (106) of the victims exhibited anaphylactic shock, which was higher than previous reported cases (1%). The high anaphylactic shock percentage was probably because most victims were aged farmers or had a higher sensitivity characteristic of Asian people. RIFA is well adapted to the environmental conditions in Taiwan, which makes it extremely difficult (if not impossible) to eradicate. The management of RIFA in the future should be focused on reducing the speed of spread to mitigate possible dangers to the inhabitants. Six major challenges of RIFA management in Taiwan are also discussed.
L26: “had a higher sensitivity characteristic of Asian people” Not sure what you mean here?
Introduction
L46-47: Reformulate this sentence please.
L50: Suggestion: “RIFA is one of the most threatening invasive species to human health because its stingers can pierce the human skin and inject venom, causing medical problems.”
L68 and L71: need a citation
L71-72: “Despite great efforts, the USA has failed to eliminate RIFA because of the lack of experience and strategic missteps.” Either cite a scientific paper proving that or remove the sentence. Or reformulate like you say in the abstract: “RIFA are almost impossible to eradicate”.
L75 and L81: replace “pandemic” by “invasion”.
L76: what is the “35%” referring to? Citations are required for this paragraph.
L87: need a citation
L92: remove “and adaptation”
Materials and methods
L95: remove “/eliminate”
L106: 1,958
L113: “of the number of people” replace “people” by “persons” and reformulate the sentence… it is not clear what you want to say
To which extent can you trust the general public’s expertise in the identification of RIFA?
L112-117: it is not clear what you tested with your stats.
Results and discussion
L125-126: need a citation
L145: need a citation
Figure 1: Text is too small
Could you plot the general public detections with one color and survey detection with another color?
L182: replace “rid” by “free”
L188: Urbanization is much more ancient than that!!!! Again, you need to cite your sources.
L188-197: The whole paragraph needs citations.
Figure 2: Reformulate the first sentence of the caption.
What is a skygarden??
F and I are just the picture of buildings, put some arrows to indicate were are located the RIFA…
Section 3.3 is mostly anecdotal. Reduce it or remove it. You don’t have data on the destruction of public facilities (just some example), that’s make your discussion longer for nothing.
L239: replace “under the BAPHIQ” by “(BAPHIQ).
L240: 10,127 (persons)
L244: it’s “Dress” according to your references, anyway, remove “According to a report by Dr. Drees [13],” and simply cite the ref at the end of your sentence.
L246: replace “epidemic” by “invaded”
L249: % not “percent”
L238-251: There is no statistic no compare your studies… Can you really compare these numbers? Maybe yes but you don’t need to spend a whole paragraph on it. The point is that people are being stung by fire ants no?
L327: the citation should be numbered.
L331: need citation
L362-364: reformulate
Table 2 should be removed; everyone can find this information on Wikipedia (for example).
References must be checked for lacking italics, uppercases, orthographic errors…
L454: “efforts” not “eorts”
Author Response

(The authors gave the same response as above.)

Reviewer 3 Report
Please read and follow the recommended corrections. n my opinion this manuscript is written like a story telling way. Therefore, you will need major changes to bring it to the scientific level as an manuscript.

Author Response

(The authors gave the same response as above.)

Reviewer 4 Report
This paper reviews the impacts of the invasive fire ant Solenopsis invicta (RIFA) on public facilities and human health in Taiwan. From my point of view the paper is well-written and informative but has some weakness and thus could still be improved.
- Information is often not supported by references.
For instance, reference(s) has to be added in paragraph written from line 70 to 76 to allow the reader to find the original source of information.
Another example : L. 231 : « The RIFA venom contains mainly alkaloids and proteins ». Could you please add reference(s) ?
Other examples could be given. Please check if all information is adequately supported by references.
- Avoid repetition of information.
For instance :
L. 41 : « The red imported fire ant (RIFA), Solenopsis invicta Buren, is classified as one of the top 100 most invasive ants worldwide »
Similar information is written a second time :
L. 79 : « RIFA is categorized as one of the world’s top 100 invasive species. »
Other examples :
L. 125 : « Taoyuan City, located in northern Taiwan, was the area where RIFA was initially discovered in 2003. »
L. 137 : « RIFA was initially discovered in northern (Taoyuan city) […] Taiwan »
Or :
L. 152 « the area of Taiwan is only 36000 km2 » and L. 307 : « However, Taiwan is a small country having an area of only 36,000 km2. »
Other examples could be added. Please check your paper and delete or reformulate when necessary to avoid this kind of repetition.
- I find the part dealing with « Destruction of public facilities by RIFA » (Lines 208-227) somehow anecdotal. Could it be possible to provide more general pattern or analyses ? For instance, could it be possible to evaluate the economic cost of RIFA in Taiwan since its introduction ?
- Meaning of data points (L. 104-107)
« A total of 34,131 data points were retrieved from the Plant Epidemic Information Network of Taiwan on March 20, 2020. Among the 34,131 cases, 27,279 (79.92%) were under treatment and control, 7 (0.02%) were in notification, 1958 (5.74%) were caused by non-invasive ants, and 4,887 (14.32%) cases were closed. »
Please could you explain what means « in notification » ? Similarly, what means « closed » ? Does it mean that RIFA was eradicated ?
- Analyses in Table 1 :
I don’t understand this table and the analyses. If the people were not stung, they have not needed medical care, they have not made wheal-and-flare reaction and so on. As a result the chi² is obviously highly significant.
As you have done in your text, you should each time restrict your group of people. L. 257 : « Of the Taiwanese people bitten by RIFA, only 21.84% (834 people) developed skin redness and swelling ». As a result, you can test if skin reaction is significant among people stung by RIFA. Same for other health issues.
By the way, prefer the verb « sting » rather than « bite » as the venom is injected by the sting apparatus, not the mandibles.
- Polygyny in RIFA
L. 368-371 : « RIFA is a social insect with the queen as the only female with fully developed ovaries responsible for the reproductivity of the whole colony. These ants can only be controlled by eradicating the queen, which usually hides inside the nest. The bait used for RIFA control can wipe out the entire colony by destroying the queen through trophallaxis. »
Are all the RIFA nests monogynous in Taiwan ? This species can indeed have both monogynous and polygynous nests. Polygyny is one of the key biological traits explaining both the invasive success of RIFA and the difficulties we may have to eradicate it. Your sentences seem to indicate that all RIFA nests are monogynous, which would be very surprising. For instance, both monogyne and polygyne social forms of S. invicta were present in China with polygyne colonies as the dominant one (Wang et al. 2020).
By the way, this key reference is missing :
Wang L, Zeng L, Xu Y, Lu Y (2020) Prevalence and management of Solenopsis invicta in China. NeoBiota 54: 89–124. https://doi.org/10.3897/neobiota.54.38584
I think it is an important one as several informative comparisons could be done.
Other issues :
- I find the title misleading as the « adaptation of RIFA » is not discussed in your paper. Perhaps « invasion » would be a better word. Also « impacts » is probably too general as environmental impact is not discussed, perhaps precise « societal impact » or something like that.
- Write scientific name in italics.
Author Response

(The authors gave the same response as above.)

Round 2
Reviewer 2 Report
I am globally satisfied with the revisions. The authors did a good job revising this manuscript. Please check the numbers associated with the references in the revised manuscript.
Maybe consider removing "strategic missteps" (line 69). It is not a fact but the authors' interpretation of Williams (2001) paper. I think that the most likely explanation of eradication failures in the US and elsewhere is that RIFA are extremely, or even nearly impossible, to eliminate once established. Given the current spread of the RIFA in Taiwan, I (sadly) guess that Taiwan will face the same invasion problem and eradication failure than the US...
Author Response
Comment 1.
I am globally satisfied with the revisions. The authors did a good job revising this manuscript. Please check the numbers associated with the references in the revised manuscript.
Response: We have revised our manuscript accordingly. All references were formatted with Endnote software and carefully check one by one.
Comment 2
Maybe consider removing "strategic missteps" (line 69). It is not a fact but the authors' interpretation of Williams (2001) paper. I think that the most likely explanation of eradication failures in the US and elsewhere is that RIFA are extremely, or even nearly impossible, to eliminate once established. Given the current spread of the RIFA in Taiwan, I (sadly) guess that Taiwan will face the same invasion problem and eradication failure than the US...
Response: We replaced ‘strategic missteps’ with ‘absence of effective natural enemies’. Moreover, this paragraph was moved to Page 8 (Line 305-306).
Reviewer 3 Report
I do not have any more comments to the Author/s
Author Response
Thanks for your review.
Reviewer 4 Report
Dear authors,
You have nicely improved your manuscript and have provided an answer to all the issues raised by the referees. I find your paper easier to read and better structured. Your text is now supported by references as requested. Your Figure 1 was also nicely improved.
However, I am still not convinced by the usefulness of the tests shown in your Table 1. My reasoning remains the same, I mean: people don't have to seek medical care if they were not stung. People will never have wheal-and-flare reactions if they were not stung (however your table indicates that 32 persons have this kind of reaction without having been stung, so I don't understand). People cannot have any anaphylactic shock if they were not stung. Right? As a result, the tests are meaningless and the chi² results are obviously highly significant. I suggest to build your table differently, without carrying out any test. Your Table S1 is more informative and perhaps it could be a good idea to put your current Table 1 in supplementary information and your Table S1 into the manuscript.
Other remarks:
- L. 65-69: I find this part on USA's eradication efforts somehow off-topic here. I understand you want to use these sentences to underline the difficulty to eradicate RIFA. However, I think your introduction should focus on the Taiwanese case. I suggest to either delete this part or to integrate it into your discussion.
- L. 180: Replace "difficulty" by "difficult".
- L. 249: Replace "In that6 case" by "In that case".
- L. 286: Replace "RIFA is likely have" by "RIFA is likely to have"
- L. 302-303: "However, at the Sungshan Airport, RIFA was wholly eradicated with a two-year extensive control regimen using a two-step method". What is this two-step method? Can you provide more details in just a few sentences, please? This would allow the reader to understand without having to read Reference 38.
- L. 317: "Taiwan’s government continues to invest significant effect to prevent its spread southward". I am not sure if the word "effect" is the best one for this sentence.
- L. 335: Replace "July; In Taiwan" by "July; in Taiwan,"
- Table S1: The label should be more precise. Perhaps write something like "Association of seeking medical care and symptomology within people having been stung by RIFA"
- Table S1: sum of column "no" is 3531, not 3541.
Author Response
Reviewer 4
Comment 1
However, I am still not convinced by the usefulness of the tests shown in your Table 1. My reasoning remains the same, I mean: people don't have to seek medical care if they were not stung. People will never have wheal-and-flare reactions if they were not stung (however your table indicates that 32 persons have this kind of reaction without having been stung, so I don't understand). People cannot have any anaphylactic shock if they were not stung. Right? As a result, the tests are meaningless and the chi² results are obviously highly significant. I suggest to build your table differently, without carrying out any test. Your Table S1 is more informative and perhaps it could be a good idea to put your current Table 1 in supplementary information and your Table S1 into the manuscript.
Response: We appreciate your valuable suggestion. We have modified and interchanged Table 1 with Table S1, and removed the chi-square test part. We have also revised our manuscripts accordingly (Page 3, Line 101-105). As to the problem: 32 persons shown in Table S1 have wheal-and-flare reaction without having been stung. We guess it should be the mis-understanding from the general public. Those residents in RIFA heavily infested area just exhibit mild wheal-and-flare reaction (e.g. psychologically), and went to a doctor even they were not stung by RIFA.
Comment 2.
- 65-69: I find this part on USA's eradication efforts somehow off-topic here. I understand you want to use these sentences to underline the difficulty to eradicate RIFA. However, I think your introduction should focus on the Taiwanese case. I suggest to either delete this part or to integrate it into your discussion.
Response: this paragraph has been moved to Page 8 (Line 304-310). Thanks for your suggestion.
Comment 3.
- 180: Replace "difficulty" by "difficult".
Response: We have revised our manuscript accordingly (Page 4, Line 171)
Comment 4.
- 249: Replace "In that6 case" by "In that case".
Response: We have revised our manuscript accordingly (Page 6, Line 236)
Comment 5.
- 286: Replace "RIFA is likely have" by "RIFA is likely to have".
Response: We have revised our manuscript accordingly (Page 7, Line 272)
Comment 6.
- 302-303: "However, at the Sungshan Airport, RIFA was wholly eradicated with a two-year extensive control regimen using a two-step method". What is this two-step method? Can you provide more details in just a few sentences, please? This would allow the reader to understand without having to read Reference 38.
Response: We have revised our manuscript accordingly (Page 8, Line 289-290)
Comment 7.
- 317: "Taiwan’s government continues to invest significant effect to prevent its spread southward". I am not sure if the word "effect" is the best one for this sentence.
Response: We have used ‘endeavor’ instead (Page 8, Line 310-311). Thanks for your suggestion.
Comment 8.
- 335: Replace "July; In Taiwan" by "July; in Taiwan,"
Response: We have revised our manuscript accordingly (Page 9, Line 328)
Comment 9.
Table S1: The label should be more precise. Perhaps write something like "Association of seeking medical care and symptomology within people having been stung by RIFA"
Response: We have revised our manuscript accordingly (Page 5, Table 1)
Comment 10.
Table S1: sum of column "no" is 3531, not 3541.
Response: We have revised our manuscript accordingly (Page 5, Table 1)
